# The Impact of Business Leaders’ Formal Health and Safety Training on the Establishment of Robust Occupational Safety and Health Management Systems: Three Studies Based on Data from Labour Inspections

**DOI:** 10.3390/ijerph19031269

**Published:** 2022-01-24

**Authors:** Øyvind Dahl, Torbjørn Rundmo, Espen Olsen

**Affiliations:** 1Department of Innovation, Leadership, and Marketing, University of Stavanger, 4021 Stavanger, Norway; oyvind.dahl@uis.no; 2Department of Psychology, Norges Teknisk-Naturvitenskapelige Universitet, 7491 Trondheim, Norway; torbjorn.rundmo@svt.ntnu.no

**Keywords:** safety and health, compliance, management systems, OSH systems

## Abstract

The impact of occupational safety and health (OSH) training is a neglected topic in safety research. In Norway, such training is mandatory for all business leaders. Hence, the Norwegian working life forms a particularly interesting case for studying the impact of OSH training. On the basis of data from labour inspections performed by the Norwegian Labour Inspection Authority (NLIA), this article examines the impact of business leaders’ mandatory OSH training on the establishment of robust OSH systems. Three separate studies have been conducted. In study 1, cross-sectional data from inspections of 29,224 companies are analysed. In study 2 and 3, longitudinal data from inspections of 1119 and 189 companies, respectively, are analysed. The analyses reveal that mandatory OSH training of business leaders is positively associated with compliance with legal requirements related to the minimum content of OSH systems. This means that mandatory OSH training is important for the establishment of robust occupational safety and health management systems.

## 1. Introduction

Accident investigations continue to illustrate how non-compliance with safety regulations plays a crucial role in the development of accidents (Hopkins, 2011). This demonstrates that adherence to safety rules and procedures, generally referred to as safety compliance, may be important in maintaining safety at work in modern workplace settings. The fact that violations with safety regulations are associated with accidents illustrates the need to identify and explain the reasons why violations occur because such knowledge can address factors that can improve safety compliance [1,2,3,4,5,6,7].

Non-compliant acts are deviations from safe operating procedures, standards or rules ([8], p. 72). Such violations can be committed not only by individual workers through violating company internal safety procedures but also by the organisation itself, through violating health and safety regulations. Dahl and Søberg [9] argue that more research is needed on such organisational non-compliance with safety regulations and also on the measures that can be taken to improve compliance at the organisational level. The majority of safety research focuses on the individual level with the aim of identifying the causes of unsafe acts and violations at the sharp end of organisations (e.g., [10,11]). Research on safety management often focuses on the relationship between leadership and individual behaviour at the sharp end of the organisation [12,13]. However, it is not well understood how managerial training influences compliance at the organisational level. Zohar [14] has shown how perceptions of safety climate and organisational factors are important. Hitherto, the majority of research in this area has focused on employees’ subjective judgements and perceptions. There is a need for more variety in approaches aimed at examining safety issues at an organisational level [15].

In this paper, the aim is to investigate the influence that managers’ OSH training has on safety compliance at the organisational level. The outcome of safety training is an under-researched topic within safety science. In particular, there is a need for more research on the relationship between training and safety compliance at the organisational level. In the paper, safety compliance at the organisational level is understood as having an occupational health and safety management system that complies with national safety regulations. Occupational safety and health management systems (OSH systems) are systematic approaches set up by employers to minimize the probability and consequence of workplace injuries and work-related illness. This paper is based on three studies of quantitative data from labour inspections performed by the Norwegian Labour Inspection Authority (NLIA).

## 2. Study Background

Most western countries have workplace health and safety regulations that incorporate and regulate OSH management. However, working environment regulations will not be effective if they are ignored and not transformed into practice. Generally, OSH initiatives have not had the impact anticipated both by experts and policymakers. The main reason for this has been a gap between policy and practice [16].

Research and theory on OSH systems have received much attention during the 1990s. The Scandinavian shaping of legislation towards the working environment [17] was influenced by the sociotechnical approach on management [18,19,20]. Today, OSH management research has become more complex, but undoubtedly, research on safety leadership and safety management systems is a core area in this type of research.

Compliance often incorporates safety leadership as particularly important for achieving adherence to rules and regulations [15]. Safety leadership research has focused on leadership styles, such as the transactional–transformational perspective. This perspective builds on both an extrinsic and intrinsic motivation approach. Intrinsic motivation is generally stimulated by transformational leadership, while transactional leadership appeals to an individual’s extrinsic motivation [15]. Transformational and transactional leadership styles have been linked to outcomes such as perceived safety climate, safety participation and safety compliance [12]. Empirical support for conclusions about the importance of leadership styles has been information that has contributed to strengthening commitment and attention to safety [21], and also, to cause priority of leadership training and development of appropriate leadership styles [12].

However, Pilbeam, Doherty, Davidson and Denyer [15] discussed the limitations related to research on safety leadership and compliance. The majority of research has been based on generic leadership scales, with a focus on supervisor–frontline worker relationships. However, other relationships in organisations have been less frequently studied. According to Pilbeam, Doherty, Davidson and Denyer [15], research should prioritize top and senior managers and examine more thoroughly how they influence safety in their own organisations.

Based on the adoption of the transactional–transformational perspective in safety leadership research, the research field does not generally include OSH system characteristics, such as the role of safety representatives, health and safety skills of leaders, application of hazard identification, risk reduction measures, and use of employment contracts. In Norwegian regulations, these factors are requirements in the Norwegian Working Environment Act (WEA) in which the employer is required to adhere to. Additionally, the employer, represented by the company managers, is required to have conducted formal OSH training. Hence, labour inspections register a level of compliance with the WEA requirements listed above. The general compliance with the WEA requirements is documented, as well if responsible managers have completed OSH training.

Saksvik et al. [22] showed that managerial OSH training was positively correlated with several OSH system characteristics, such as training of employees, a cooperative implementation strategy, assessments, actions plans, OSH improvement actions, interest of management and interest of employees. These results indicated that managerial OSH training is an important factor for developing high-quality OSH systems. A review study conducted by Robson et al. [23] particularly addressed whether OSH training had beneficial effects related to workers. The findings revealed that OSH training distributed to employees was positively associated with worker practices. However, this review did not find strong relations between OSH training and health (i.e., symptoms, injuries, illnesses), and the study did not look into the effects of managerial OSH training.

Although the research suggests that managerial OSH training may have positive effects on OSH system characteristics [22], training targeting the first line supervisions level is more common than the training of top management [24]. According to Hale, Guldenmund, Van Loenhout and Oh [24], “the top management, despite what it may think about itself, does not have the clear vision, motivation and knowledge of what to achieve and how to do it, unless given this sort of input. Our evidence suggests that, if given it, they can fulfil their motivating roles much better to energise the interventions.” Moreover, top management also has the authority to take higher level decisions, e.g., those related to investments in OSH systems. Managers without OSH training or competence might down-prioritize safety representatives, and other requirements in the WEA will also require some time and resources for OSH matters. Lack of managerial OSH training can be negatively related to organisational level compliance of the WEA.

### Managerial OSH Training and Organisational Compliance

The research shows that a large number of organisations breach OSH regulations, and according to [25], there is reason to believe this is a challenge. Moreover, “a recurring conclusion in accident investigations and analyses is that non-compliance with safety regulations is a contributory factor to accidents” ([9] p. 1). The current paper explores how managerial OSH training relates to compliance of the Norwegian WEA. Implementation of OSH regulations enhances the probability of an increase in the number of formal labour inspections that can monitor and register compliance levels. Weissbrodt and Giauque [26] showed that inspections by labour inspectors can be associated with lower psychosocial risks. However, the effectiveness of workplace visits was moderated by several situational and organisational factors, implicating that more research is needed within the field [26]. Fernández-Muñiz et al. [27] illustrated that safety compliance depended on the level of worker participation in safety activities. Hence, encouragement of workers to participate actively in the area of health and safety is recommended. Moreover, since managers may influence OSH systems, managerial OSH training might increase attention towards the value of the OSH system, which might increase investments, prioritization and integration of OSH in organisations.

In the current study, we draw attention to formal OSH training of managers and use labour inspection data at the organisational level to investigate how managerial OSH training might be positively related to OSH standards. To be explicit, the outcome measure in the current study is in compliance with Norwegian WEA regulations, which is measured in accordance with labour inspections. We expect that OSH training of managers will be positively associated with compliance. The majority of studies that has confirmed the link between training and safety compliance are carried out at the individual level (e.g., [28,29]). Far fewer studies have been carried out to explore this link at the organisational level. However, a recent study [30] among small- and medium-sized enterprises has shown a significant impact of safety training of OSH coordinators on organisational safety compliance. By auditing the companies involved in the trainings study, the researchers found that the applied organisational compliance score increased by 73% after safety training.

However, the effects of OSH training might be substantially restrained by different barriers to OSH interventions. Masi and Cagno [31] conducted a study indicating that the ten most frequent barriers for OSH interventions were related to three main problem areas: regulation, resources and information. These barriers were mainly concentrated in the design and implementation phases of OSH interventions and were more frequently found among micro- and small enterprises, and less among medium-sized enterprises. Four of the barriers found were related to management flaws. These were systematically wrong behaviour of management, management not adequately skilled, lack of knowledge of the criticality by the management, and lack of knowledge of the profitability of the intervention by the management. Considering these barriers, managerial OSH training might increase OSH competence and awareness, which in turn might reduce the likelihood of OSH barriers. OSH management challenges are also more common in smaller organisations [31], and therefore we use company size as a control variable in study 1.

According to the literature presented above, the current study is based on the basic assumption that there is a positive relationship between mandatory OSH training and the establishment of robust OSH systems. This basic assumption is explored in three studies by the following three hypotheses:

**Hypothesis** **1** **(study 1):**Mandatory OSH training of managers has a positive impact on the establishment of robust OSH systems, such that companies where the manager has undergone training will comply more with the regulations compared to companies where the manager lacks training.

**Hypothesis** **2** **(study 2):**Companies having managers with mandatory OSH training at T1 (first inspection) will not improve their OSH system at T2 (second inspection), i.e., compliance with the health and safety regulations will not improve from T1 to T2.

**Hypothesis** **3** **(study 3):**Managers will improve their company’s OSH system subsequent to accomplishing mandatory OSH training, such that compliance at T2 (post-training) will be greater than at T1 (pre-training).

## 3. Methods

### 3.1. Data

This article is based on three studies which all apply data derived from regular health and safety inspections performed by the Norwegian Labour Inspection Authority (NLIA). The inspections in all three studies are performed over a four-year period. The NLIA is a government agency under the authority of the Ministry of Labour and Social Affairs. The agency has approximately 600 employees and performs roughly 15,000 inspections per year [32]. The overall mission of the NLIA is to ensure safe and healthy work environments by assuring that the enterprises comply with the existing OSH regulations. The main instrument applied by the agency is on-site health and safety inspections of companies. The aim of inspections is to control and enforce compliance with the national OSH regulations.

As shown in Table 1, study 1 included data gathered from health and safety inspections of 29,224 companies. The majority of the sample consisted of companies sized between 0 and 19 employees (66.8%). Furthermore, companies within wholesale and retail trade (22.8%) constituted the largest type of industry, followed by construction (15.6%) and accommodation and food service (12.7%).

Studies 2 and 3 included 1119 and 189 companies, respectively. Unlike in study 1, however, each of these companies was inspected twice (at T1 and T2). Thus, the number of inspections included were 2238 in study 2 and 378 in study 3. The majority of the sample in study 2 consisted of companies sized over 19 employees (60.8%). Moreover, companies within manufacturing (19.2%) constituted the largest type of industry, followed by construction (17.3%) and wholesale and retail trade (16.2%). In study 3, the majority of the sample consisted of companies sized between 0 and 19 employees (66.1%.). With regard to type of industry, accommodation and food service (30.2%) constituted the largest group, followed by wholesale and retail trade (21.2%) and construction (15.9%).

### 3.2. Measures and Descriptive Statistics

During all NLIA inspections, the health and safety inspector applies a predefined checklist. The use of checklists allows for easier examination to identify hazards in the workplace and breaches of the regulations. Typically, each checkpoint in the list represents a statutory requirement. The content of a checklist and the number of checkpoints included in the checklist vary from one inspection to another, depending on the type of industry and the focus of the inspection. For example, checklists applied at a construction site will be quite different from those applied at a repair shop, and checklists applied to uncover problems related to chemical exposure will be quite different from those related to the prevention of slips, trips and falls. Despite this difference, there are some general checkpoints that are applied frequently across different industries and across different inspection topics. This allows for studies of large sample sizes. When such checkpoints are applied twice in the same company, it also allows for comparison over time.

In the three studies underlying this article, seven checkpoints were found to be applied frequently enough to be included in the study; see Table 2. The first checkpoint (#1 in Table 2) is related to mandatory OSH training, i.e., the inspector controls whether the employer has undergone training or not. The remaining 6 checkpoints (#2 to #7 in Table 2) capture compliance with essential OSH system requirements.

During an inspection, all checkpoints are either assigned a “yes” (value 1), indicating compliance, or a “no” (value 0), indicating non-compliance. Thus, the mean value of a given checkpoint will theoretically vary between 0 (indicating that none of the inspected companies comply with the given statutory requirement) and 1 (indicating that all of the inspected companies comply with the given statutory requirement).

As described, all checkpoints included in the three studies represent a statutory requirement. Thus, if a company is assigned a “no” on a given checkpoint, this will also trigger a formal order. Hence, the potential for systematic errors, bias and reduced validity were reduced in all three studies by the fact that a “no” would trigger a juridical consequence.

Mandatory OSH training was measured by checkpoint #1, such that companies where the manager had undergone training were given the value 1, otherwise the value 0. Thus, the mean value of 0.81 on checkpoint 1 in study 1 shows that 81% of the inspected companies satisfies the requirement of mandatory OSH training.

Compliance with health and safety regulations was measured by an index (see Table 2) which captured each company’s mean score on checkpoint #2 to #7. In cases where not all these six checkpoints were controlled, the mean value of the checkpoints actually controlled was calculated. Thus, a given company’s score on the compliance index was calculated by the following formula:$$compliance\;index=\frac{checkpoints\;satisfied}{checkpoints\;controlled}$$

### 3.3. Statistical Procedures

#### 3.3.1. Study 1

In study 1, we used hierarchical multiple linear regression analysis [33] to test the hypothesized positive relationship between mandatory OSH training and the establishment of robust OSH systems (Hypothesis 1). The hierarchical procedure means that company size was entered as a control variable in the first step and OSH training in the second step. Company size was chosen as a control variable because previous studies have shown a positive relationship between size and compliance [34]. Assessment of the Hypothesis was based on a *p* value and the direction of the regression coefficient (B). The significance level was set at á = 0.05. The standardized regression coefficient (â) was used to assess the effect of training relative to company size. The model as a whole was evaluated in terms of percentage of variance accounted for (R2) and the improvement in variance accounted for between steps 1 and 2 (ΔR2). In addition to the regression analysis, we examined the bivariate relationship between training and compliance by use of a two-tailed independent samples t test. For this test, we split the sample into two sub-samples: (1) companies where the manager has undergone training and (2) companies where the manager lacks training. The significance level for the t test was set at 0.05. The effect size was estimated by Cohen’s d in study 1, as well as in 2 and 3. According to Cohen’s [35] guideline for interpreting d, a small effect produces a Cohen’s d of 0.20, a medium effect produces a Cohen’s d of 0.50, and a large effect produces a Cohen’s d of 0.80 or greater.

#### 3.3.2. Study 2

In study 2, we used a two-tailed paired sample ttest to test the assumption that companies with managers with mandatory OSH training at the first inspection will not have improved their OSH system at the second inspection (Hypothesis 2). The paired sample *t*-test (sometimes referred to as “dependent means *t*-test” or “matched-pairs *t*-test”) is a procedure used to determine whether the mean difference between two pairs of observations is significantly different from zero [36]. The significance level for the *t*-test was set at 0.05.

#### 3.3.3. Study 3

Similar to study 2, a two-tailed paired sample *t*-test was performed in study 3. The *t*-test tested the assumption that companies with managers but without mandatory OSH training at the first inspection significantly improve their OSH system at the second inspection (Hypothesis 3). Similar to study 2, the significance level was set at 0.05.

## 4. Results

### 4.1. Study 1

The results from the independent sample *t*-test are presented in Table 3. As shown in the table, companies where the manager has undergone OSH training score significantly higher on each checkpoint compared to companies where the manager lacks OSH training. For example, the no-training group’s mean score on the hazard identification checkpoint is 0.44, whereas the training group’s mean score on the same checkpoint is 0.74 (a mean difference of 0.30 and a percentage difference of 68%). This means that within the no-training group, only 44% of the companies have identified hazards and, on this basis, assessed the risk of injury or danger to workers’ health and safety, whereas the same is true for 74% of the companies within the training group. Moreover, the *t*-test which concerns the overall compliance index shows that companies where the manager has undergone OSH training (M = 0.88, SD = 0.22) comply significantly more with the health and safety regulations compared to companies where the manager lacks training (M = 0.64, SD = 0.35), t (29,222) = 62.098, *p* < 0.001. Thus, the *t*-test supports Hypothesis 1. Moreover, the effect size measure of Cohen’s d was 0.79. This roughly corresponds to a large effect [35].

The results from the regression analysis are presented in Table 4. The regression analysis was conducted in two steps, with company size added in model 1 and OSH training added in model 2. This allows for the examination of the effect of OSH training controlled for company size.

As seen in model 1, company size was positively related to compliance with the health and safety regulations. With the exception of companies sized between one and four employees, all groups scored significantly higher than the reference category (i.e., companies without employees). The results also indicate a linear increase, i.e., the larger the company, the more compliant with the regulations. In summary, company size alone accounted for 5.0% of the variance in the compliance index.

Introducing OSH training in model 2 led to a substantial increase in variance accounted for. The explained variance (R2) increased significantly from 5.0% to 13.9%. As shown in the table, OSH training is positively and significantly related to the compliance index. Hence, similar to the bivariate *t*-test, the regression analysis supports Hypothesis 1 and shows that the effect of OSH training is significant even when controlled for company size. The B-value shows that a one-unit increase (i.e., from no training to training) results in a 0.212 unit increase in the compliance index. Furthermore, the standardized regression coefficient (â) shows that the effect of OSH training (0.308) is far more powerful than the effect of any of the company size variables.

### 4.2. Study 2

The results from the paired sample *t*-test are presented in Table 5. The number of pairs differs between each checkpoint. As described in Section 3.2, this is due to the fact that not all of the six checkpoints were controlled in all inspections. Thus, a given company’s score on the compliance index was calculated by taking into consideration the number of checkpoints actually controlled.

As shown in the table, companies with managers with mandatory health and safety training at both T1 (first inspection) and T2 (second inspection) scored significantly higher on the compliance index at T2 compared to T1; t (1118) = 3.310, *p* < 0.001. This finding does not support Hypothesis 2, which stated that companies with managers with OSH training at T1 (first inspection) will not improve their compliance with the health and safety regulations at T2 (second inspection). Although significant, the increase in the compliance index from T1 to T2 was only 3.37%. Furthermore, only one of the six individual checkpoints (safety representative) showed a significant increase. Moreover, the effect size measure of Cohen’s d was 0.10. This is lower than Cohen’s threshold for small effects (i.e., 0.20) and corresponds to what Cohen refers to as a “trivial” effect (Cohen, 1988). Thus, the observed difference in compliance between the first and the second inspection is significant but trivial for companies led by managers with OSH training at T1. Since the compliance level was high and relatively stable at both inspection periods, Hypothesis 2 was partially supported, although some minor improvement of compliance was observed.

### 4.3. Study 3

The results from the paired sample *t*-test of Hypothesis 3 are presented in Table 6. As shown in the table, companies led by managers that have not undergone mandatory OSH training at T1 (first inspection) scored significantly higher on the compliance index at T2 (second inspection) compared to T1, t (188) = 6.284, *p* < 0.001. This finding supports Hypothesis 3, which stated that managers will improve their company’s OSH system subsequent to accomplishing mandatory OSH training. In contrast to the findings in study 2, where the increase in the compliance index from T1 to T2 was only 3.37%, the increase from T1 to T2 for this group of companies was 29.85%. In spite of the lower sample size, three of the six individual checkpoints showed a significant increase. Moreover, the effect size measure of Cohen’s d was 0.46. This corresponds roughly to Cohen’s [35] threshold for medium effects (i.e., 0.50). Thus, the observed difference in compliance between the first and the second inspection is significant and roughly medium sized for companies led by managers who undergo mandatory OSH training in the period between the two inspections.

## 5. Discussion

Within the context of relatively limited research on the potential benefits of managerial OSH training [15], this is the first study to report that managerial OSH training is positively related to the establishment of robust OSH systems across branches in Norway. In the first study, data from labour inspections illustrated that managerial OSH training was positively related to OSH compliance; companies where the manager had undergone formal OSH training could document compliance when visited by labour inspectors. The first study also revealed that size matters; larger organisations consistently report higher compliance compared to smaller organisations. Hence, Hypothesis 1 was supported and the effect of managerial OSH training was significant even when controlling for the size of organisations. Longitudinal data from study 2 indicated that when managerial OSH training had taken place at both the first (T1) and second inspections (T2), the compliance level was high and relatively stable during both inspection periods. Hence, the findings from study 2 gave some level of support to Hypothesis 2. Study 3 confirmed the positive effect of managerial OSH training; the level on the compliance index increased, with 29.85% based on implementation of managerial OSH training. Hence, study 3 supported Hypothesis 2 and illustrates that organisations which had not completed managerial OSH training at T1 were able to improve their OSH system considerably at T2 when the managers had completed OSH training.

The results of studies 1 and 3 illustrate that managerial OSH training is positively linked to OSH compliance. Hence, the results implicate that business leaders’ formal health and safety training has a significant impact on the establishment of robust occupational safety and health management systems (SMS). OSH training seems to extend the managerial role beyond focus on basic business topics such as marketing, production and efficiency. OSH training can create new or increased managerial consciousness of several OSH issues that are important to monitor, maintain or to develop SMS. This can for instance be related to new competencies, values, priorities and assessments related to several areas, such as (1) assessment of work hazards, (2) development of preventive measures, (3) establishment of OSH related roles such as safety deputies, (4) participation and involvement of employees, (5) limitations and shortcomings with regard to OSH competencies and current state of the organisation, and (6) importance of monitoring and controlling work load and maximum working hours. Many of these areas are fundamental elements of OSH systems (e.g., [37]) and may also be specifically defined requirements in labour laws and regulations. Moreover, if managerial OSH training changes the priorities and values of top managers, this can explain why managerial OSH training is associated with increased compliance across organisations. This influence can for instance include investments and attention towards further improvement of OSH system elements in order to improve performance related to health and safety. Another potential is that OSH training improves manager commitment towards safety, which in turn increases and enforces safety awareness and safety priorities across organisations. A key to safety is on-going mindfulness about the possibility of failure, detection of early warnings systems, and learning from incidents (Weick, 1999). As such, safe organisations constantly attend to their safety management system and risk control and adapt their systems to new hazards, insight and conditions (Hopkins and Hale, 2002).

The results from study 2 complement the results from studies 1 and 2. While studies 1 and 2 demonstrate that managerial OSH training is associated with increased compliance across organisations, study 2 indicates that the compliance level is relatively stable when managers have conducted OSH training at times 1 and 2. This finding implies that repeated managerial OHS training will potentially have little effect on compliance. In addition, study 2 indicates that the managerial OSH training should have different formats over time to benefit the organisations’ potentiality. Other training interventions could for instance focus more on creating improvements related to safety behaviour and safety culture in organisations [38]. Another issue is the complexity of measures and the length of improvement programmes. Empirical research illustrates the difficulties of improving safety culture quickly, and that safety intervention could build on more refined theoretical models [39]. These issues should also be considered when developing OSH measures in organisations. It is possible to focus both on bottom-up and top-down processes when improving OHS [38], and designers of safety programmes should consider both approaches. Another issue is not to consider OSH in isolation but to consider human resource perspectives in parallel. Research indicates that multiple general issues are related to OSH and are essential to consider, such as remote work during the pandemic [40], work climate [41], psychological needs of workers [42], change processes [43], cognitive functions [44], and meaning of work [45]. These topics are essential in addition to managerial OSH training.

The results also indicate that organisation size matters for the quality of the safety management systems; larger organisations are consistently associated with better scores in OSH compliance. This may be related to the natural evolvement of organisations; larger organisations most often have a longer history and have had more opportunities and resources to build government systems, conduct training, and recruit competent personnel. The results indicate that smaller organisations with less employees probably are associated with poorer OSH management characteristics and increased risk of non-compliance with OSH regulations. Hence, labour inspection organisations might increase their attention towards smaller organisations because these are more likely to not comply with standards, which again is possibly associated with higher risk.

Traditional safety leadership research does not look at formal OSH competence and leadership training. This study expands much of the traditional research linking transformational–transactional leadership with compliance by investing in how managerial OSH training is related to compliance across organisations. We found positive and consistent associations between managerial OSH training and establishment of well-functioning OSH systems (compliance). The quality of managerial OSH training was not assessed as part of the labour inspections but only if formal managerial OSH training could be documented as part of the inspections. Hence, we did not compare the relative effectiveness of different OSH training methods [46] and did not control for the quality level of OSH training completed by the managers. In Norway, regulations do not set a requirement for the standards of formal managerial OSH training, only that formal OSH training must have been completed by the responsible managers within companies. However, in Denmark, regulations require that vendors offering OSH training to companies are certified and formally approved to satisfy specific regulatory requirements. Hence, it is possible to presume that formal OSH training is better controlled with regard to quality and content in Denmark in comparison to Norway. Nevertheless, formal OSH training of managers is clearly associated with better compliance levels across Norwegian companies. We, therefore, believe that there might be a potential for an even higher influence of formal OSH training, e.g., if the OSH courses could be improved through a form of quality assurance or certification. This study can therefore be replicated, for instance in Denmark, to observe the effect of formal OSH training that has been certified to see if the association with compliance might be even stronger than for the current study conducted in Norway.

## 6. Limitations, Future Research and Conclusions

This paper revealed that mandatory OSH training of business leaders is positively associated with compliance with legal requirements related to the minimum content of OSH systems. This means that mandatory OSH training is important for the establishment of robust occupational safety and health management systems.

Earlier, studies indicated that labour inspections can improve OSH systems [9]. This study continues further because managerial OSH training and other compliance indicators are collected and assessed in relation to labour inspections. The current study revealed that organisations with lower levels of managerial OSH training are clearly associated with lower OSH compliance. However, in the follow-up inspections (study 3), when managers had undergone OSH training at T2, scores on compliance indicated substantial improvement from the first inspections (T1). Hence, results show that managerial OSH training has a substantial influence on OSH systems. Still, it is not possible to exclude the possibility that a part of the improvement might be related to expectations or investments associated with preparing for forthcoming labour inspections. Some caution should therefore be considered in the interpretation of the results, and future studies need to look at the unique or combined influence of labour inspections, managerial OSH training and OSH system improvements. Another topic is the sustainability of the different improvement efforts in relation to compliance initiatives, which also needs to be addressed in future research.

Managerial and leader competence is generally not heavily emphasized in scales measuring transformational leadership behaviour. The results presented in this paper suggest that OSH training and knowledge deserve more attention in safety leadership research. Why is the competence of managers and leaders not explicitly defined in transformational leadership scales? Is it reasonable to believe that managers and leaders can inspire followers with regard to OSH challenges without certain levels of OSH training and formal competence? Can managers and leaders conduct individualized considerations regarding safety and risk without safety understanding? Is it not likely that leaders lacking OSH competence will find it harder to intellectually stimulate followers related to OSH challenges? Based on these results, forthcoming research can investigate these issues and how OSH competence is related to managerial and leadership roles as well as OSH standards and OSH performance of organisations. Equally important, the results clearly indicate that OSH training contributes to more compliance at the organisational level. The results show that this is not only visible between companies (study 1) but also within companies (study 3) when previously untrained managers undergo training. This shows the importance of making such training requirements mandatory and not voluntary.

Most studies on safety compliance have studied safety compliance at the individual level [15], whereas few studies have focused on identifying effective measures at the organisational level across industries. Future studies can build on the current study to further investigate compliance and OSH systems at the organisational level, potentially in combination with intervention designs, which are still limited and are needed within safety research [47].

## Figures and Tables

**Table 1 ijerph-19-01269-t001:** Descriptive statistics for companies.

	Study 1		Study 2		Study 3	
Characteristics	Frequency	Percent	Frequency	Percent	Frequency	Percent
**Company size**						
0	1207	4.1	12	1.1	3	1.6
1–4	4320	14.8	40	3.6	17	9.0
5–9	6884	23.6	129	11.5	46	24.3
10–19	7110	24.3	257	23.0	59	31.2
20–49	5630	19.3	302	27.0	46	24.3
50–99	2025	6.9	159	14.2	9	4.8
100–249	1278	4.4	140	12.5	7	3.7
≥250	770	2.6	80	7.1	2	1.1
**Type of industry**						
Wholesale and retail trade; repair of motor vehicles	6674	22.8	181	16.2	40	21.2
Construction	4555	15.6	194	17.3	30	15.9
Accommodation and food service activities	3722	12.7	142	12.7	57	30.2
Manufacturing	3072	10.5	215	19.2	25	13.2
Human health and social work activities	2492	8.5	67	6.0	5	2.6
Administrative and support service activities	1510	5.2	69	6.2	3	1.6
Transportation and storage	1325	4.5	32	2.9	6	3.2
Other	5874	20.2	219	19.5	23	12.1
**Total**	29,224	100.0	1 119	100.0	189	100.0

**Table 2 ijerph-19-01269-t002:** Descriptive statistics for compliance index and checkpoints in studies 1, 2 and 3.

			Study 1		Study 2		Study 3	
#	Topic	Checkpoint	Mean	SD	Mean	SD	Mean	SD
1	OSH training	Has the employer undergone training in health and safety work?	0.81	0.39	1.00	0.00	0.50	0.50
2	Safety representative	Has the company chosen a safety representative?	0.76	0.43	0.90	0.29	0.69	0.46
3	Employment contract	Has a written employment contract been signed with the employees?	0.92	0.27	0.97	0.18	0.85	0.35
4	Health and safety skills	Do employees assigned to lead others have the necessary skills such that the consideration of safety and health is taken care of within the manager’s area of responsibility?	0.94	0.24	0.95	0.22	0.89	0.32
5	Working time overview	Does the employer have a continuous overview of how much the individual employee works?	0.92	0.27	0.95	0.21	0.85	0.35
6	Hazard identification	Has the employer identified the hazards and problems the employees may be exposed to in the business and, on this basis, assessed the risk of injury or danger to workers’ health and safety?	0.70	0.46	0.80	0.40	0.68	0.47
7	Risk reduction measures	Has the employer taken the necessary measures and/or plans that describe measures to remove or reduce hazards and problems at work?	0.74	0.44	0.82	0.38	0.73	0.44
	Compliance index		0.83	0.27	0.90	0.20	0.77	0.33

**Table 3 ijerph-19-01269-t003:** Scores on checkpoints and index (no training vs. training), including *t*-test of mean difference.

OSH Training	No Training		Training		Difference		
Topic	Mean	SD	Mean	SD	Mean Diff.	% Diff.	
*Compliance index*	*0.64*	*0.35*	*0.88*	*0.22*	*0.23*	*37.50%*	**
Safety representative	0.43	0.50	0.82	0.38	0.39	90.70%	*
Employment contract	0.79	0.41	0.96	0.20	0.17	21.52%	*
Health and safety skills	0.81	0.39	0.96	0.21	0.15	18.52%	*
Working time overview	0.82	0.38	0.95	0.23	0.13	15.85%	*
Hazard identification	0.44	0.50	0.74	0.44	0.30	68.18%	*
Risk reduction measures	0.54	0.50	0.77	0.42	0.23	42.59%	*

* *p* < 0.001. ** *t*-test: df = 29,222, t = 62.098, *p* < 0.001 (effect size: Cohen’s d = 0.79), italic means the sum of the others below.

**Table 4 ijerph-19-01269-t004:** Linear regression with compliance index as a dependent variable. Including unstandardized (B) and standardized (β) regression coefficients.

	Model 1			Model 2		
Characteristics	B	β	*p*	B	β	*p*
Constant	0.753		*	0.613		*
1–4 employees	0.015	0.019		0.013	0.017	
5–9 employees	0.019	0.029	**	0.000	0.000	
10–19 employees	0.086	0.136	*	0.050	0.080	*
20–49 employees	0.138	0.201	*	0.089	0.130	*
50–99 employees	0.159	0.150	*	0.104	0.098	*
100–249 employees	0.181	0.137	*	0.119	0.090	*
≥250 employees	0.192	0.114	*	0.129	0.076	*
OSH training				0.212	0.308	*
F-value (∆)	219.35		*	3037.14		*
R^2^	0.050			0.139		
∆R^2^	0.050			0.089		

* *p* < 0.001. ** *p* < 0.05.

**Table 5 ijerph-19-01269-t005:** Scores for compliance index and checkpoints for companies with OSH training at both T1 and T2 (study 2). Including paired samples *t*-test of mean difference.

Topics	Pairs	Mean T1	Mean T2	Mean Diff.	% Diff.	
*Compliance index*	*1119*	*0.89*	*0.92*	*0.03*	*3.37%*	**
Safety representative	847	0.89	0.95	0.06	6.74%	*
Employment contract	438	0.96	0.96	0.00	0.00%	
Health and safety skills	587	0.96	0.95	−0.01	−1.04%	
Working time overview	273	0.92	0.95	0.03	3.26%	
Hazard identification	347	0.80	0.83	0.03	3.75%	
Risk reduction measures	324	0.83	0.82	−0.01	−1.20%	

* *p* < 0.001. ** *t*-test: df = 1118, t = 3.310, *p* < 0.001 (effect size: Cohen’s d = 0.10), italic means the sum of the others below.

**Table 6 ijerph-19-01269-t006:** Scores for compliance index and checkpoints for companies with managers that lack OSH training at T1 but who have undergone OSH training at T2 (Study 3), including paired sample *t*-test.

Topic	Pairs	Mean T1	Mean T2	Mean Diff.	% Diff.	
*Compliance index*	*189*	*0.67*	*0.87*	*0.20*	*29.85%*	***
Safety representative	115	0.64	0.91	0.27	42.19%	*
Employment contract	111	0.82	0.91	0.09	10.98%	**
Health and safety skills	41	0.78	0.98	0.20	25.64%	**
Working time overview	71	0.82	0.86	0.04	4.88%	
Hazard identification	31	0.77	0.84	0.07	9.09%	
Risk reduction measures	30	0.83	0.87	0.04	4.82%	

* *p* < 0.001. ** *p* < 0.05. *** *t*-test: df = 188, t = 6.284, *p* < 0.001 (effect size: Cohen’s d = 0.46), italic means the sum of the others below.

## Data Availability

The data presented in this study are available on request from the corresponding author.

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
