# Peer review of "The Impact of Business Leaders’ Formal Health and Safety Training on the Establishment of Robust Occupational Safety and Health Management Systems: Three Studies Based on Data from Labour Inspections"

_ijerph, 2022, doi:10.3390/ijerph19031269_

Round 1
Reviewer 1 Report
Dear authors, I think you deal with an interesting topic and I am pleased about that.
Nevertheless, I frankly find your study of no particular interest for publication in a scientific journal because it does not, in essence, add any knowledge to the scientific literature on occupational safety.
Your article is, in general, a pleasant reading. In spite of this, it is clear that the intentions and tones that you sometimes state when, for example, you talk about the importance of the organisational climate for safety, or of safety leadership (topics of great importance for the psychology of organisations and occupational medicine!) do not find particular support in the more modest analyses that you carry out.
Basically, behind the big label of "3 studies" and "longitudinal data in 2 of them" you emphasise the fact that those who have taken the safety course are more inclined to establish important safety policies, that there are no changes in the type in companies with managers who have already complied with this requirement, and that instead those who take the course (which is compulsory, so why are there those who have not taken it?) tend after some time to strengthen safety in their organisation. Also, you find better results for large companies.
Even without considering the rather hasty way of drafting the paper, the somewhat bizarre formulation of hypotheses disconnected from the supporting literature (interesting but directly supporting your hypotheses?), the somewhat too emphatic tone, the "generous" division into as many as 3 studies, I mean... My question is: do you really need a scientific article to "prove" this? Is this what our discipline is being reduced to? Forgive me, but there is a risk of "discovering hot water".
I would like to point out that I am not in the least questioning the truthfulness of what you say, or the interestingness of the results, but the scientific nature of this information, which is perhaps more useful for a report to the legal authorities of your country than an enrichment of the scientific literature on the subject of safety at work.
Forgive me, but this is my opinion.
Sincerely
Author Response
Thank you for your honest review and comments and for taking the time to consider the paper.
We developed this paper based on unique datasets, and we honestly believe this paper make an outstanding contribution to the field and that the article deserves publication in a high impact quality journal like IJERPH. We are, therefore, sorry to read that you have a somewhat critical perspective on this paper, which generally contradicts the majority of the other reviewers. We shed some light on different approaches used within the field (e.g. org. climate and perception studies) to illustrate the research gap and what this study adds.
Even though safety training is regulated by law, the compliance of regulation issues will always vary between companies of different sizes resulting in high risk. We think it is an absolute necessity to study and develop empirical evidence on the hypotheses that we have developed (and we definitely believe it is necessary to "prove" and create empirical proof).
We also think the paper is very scientific. Since most studies within the field use perception questionnaires and latent factors, this study is fascinating since it is based on actual observation within and across companies and that very robust samples are gathered based on the method employed (labour inspections).
Reviewer 2 Report
Add publications of current articles on the subject of this article
Author Response
Thank you for your positive review. It is highly appreciated. We have added some new references.
Reviewer 3 Report
The paper has some flows that need to be revised.
(1) The significance and the newness of your study is still not clear even you tried to present the previous studies.
(2) There is no concrete theoretical framework regarding the factors? The arguments are too shallow, it lacks substance to support the hypothesis. Regarding hypotheses, there is not enough references to support your hypothesis
(3) The arguments for the hypothesis were not clear, in relations to previous and current research.
(4) The reference are too old and need more detailed and updated journals in relations to your studies.
(5) No Table provides the supporting references for the items to measure the constructs.
(6) The paper only use linear regression and t-test, the paper should use statistical method such as SEM
(7) The conclusion part provides only a brief summary of the discussion without clear contribution to knowledge. Doesn’t have strong practical and research implications.
(8) I think the authors should explain and answer in concise manner the questions they posted in their research
(9) The references in the paper are not properly formatted.
Based on all of this, I think your manuscript is not publishable in its current format. I am recommending you make major revisions to your work in order to be accepted. I hope you find my comments useful.
Author Response
Thank you for you thorough review and comments, we highly appreciate this. Please find our comments and responses below.
- The significance and the newness of your study is still not clear even you tried to present the previous studies.
RESPONS:
The significance of the study is now made clearer in part 1.0.
- There is no concrete theoretical framework regarding the factors? The arguments are too shallow, it lacks substance to support the hypothesis. Regarding hypotheses, there is not enough references to support your hypothesis
RESPONS:
In the first version of the paper, we believe that we jumped to quickly to the hypotheses without summing up the basic assumptions made from the theoretical contributions and previous research. We have now made a quick summary before presenting the hypotheses, and believe this now makes this point clearer.
- The arguments for the hypothesis were not clear, in relations to previous and current research.
RESPONS:
See response above (point 2).
- The reference are too old and need more detailed and updated journals in relations to your studies.
RESPONS:
Thank you. New and updated references have been inserted.
- No Table provides the supporting references for the items to measure the constructs.
RESPONS:
As explained in part 3.2, the measures are based on predefined checklists applied by health and safety inspectors. These checklists are based on legal requirements set out in the health and safety regulations. As we see it, it is therefore not relevant to present any further references on this.
- The paper only use linear regression and t-test, the paper should use statistical method such as SEM
RESPONS:
We do not include latent measures in this study since observations are based on labour inspections. Hence, we did for instance, not need to perform confirmatory factor analyses, and there was, therefore, no need to conduct structural equations modelling to test hypothesis.
- The conclusion part provides only a brief summary of the discussion without clear contribution to knowledge. Doesn't have substantial practical and research implications.
RESPONS:
We do agree that the conclusion part focus more on theory and research, without being clear on the practical implications of the study. We have now added the following:
Equally important, the results clearly indicate that OSH-training actually do contribute to more compliance at the organizational level. The results show that this is not only visible between companies (study 1), but also within companies (study 3) – when pre-viously untrained managers undergo training. This shows the importance of making such training requirements mandatory and not voluntary.
- I think the authors should explain and answer in concise manner the questions they posted in their research
RESPONS:
Thank you for this comment. The following text was inserted in part 6.
This paper and analyses reveal that mandatory OSH-training of business leaders is positively associated with compliance with legal requirements related to the minimum content of OSH-systems. This means that mandatory OSH-training is important for the establishment of robust occupational safety and health management systems.”
- The references in the paper are not properly formatted.
RESPONS:
Thank you. We did our best to comply with the journal guidelines.
We updated the reference style in part 2 related to two places/references (from line 63).
We also downloaded an used the MDPI.references style file in EndNote. We also hope the editorial office will potentially improve the reference list if there still are some concerns.
Reviewer 4 Report
Congratulations on preparing an interesting paper based on extensive data.
Author Response
Thank you. Your positive review is highly appreciated.
Reviewer 5 Report
Dear authors,
I congratulate you on your work. It is an interesting approach to the competencies needed for leadership, focusing on something that is unusual.
Kind regards,
IPR
Author Response
Thank you. We honestly believe this is an innovative and unusual contribution within the field, and your positive review is highly appreciated.
Round 2
Reviewer 1 Report
Dear authors,
The few changes you have made have certainly improved the clarity of the paper, but in my opinion it suffers from the same weaknesses already highlighted in my previous report.
I appreciate your responses, which certainly highlight strengths of the study that I myself had observed in the previous review of this paper.
However, I believe they are insufficient and far more minor than the weaknesses already presented.
Since there are no significant improvements in this sense, I can only confirm, albeit with regret, my judgement.
I wish you all good work.
Sincerely
Author Response
Thank you for wishing us good work and for taking the time to review our study which we still think is very original and based on an extensive amount of empirical data.
Fortunately, the majority of reviewers are satisfied with this study.
We wish you all the best.
Reviewer 3 Report
Thank you so much for revising your paper but I believe, you still need to revise move to enhance the quality of your paper.
1. For example, the introduction part, this whole paragraph was copy paste from previous paper: "Non-compliant acts, defined as deviations from safe operating procedures, standards or rules [8, p. 72] can take place both at the individual level, in relation to company internal safety procedures, and at the organisational level, in relation to national occupational 36 health and safety regulations. Dahl and Søberg [9] argue that more research is needed on 37 organizational compliance or non-compliance with safety regulations and the measures 38 that can be taken to improve compliance at the organizational level." Try checking in Turnitin before you submit your revision, to double check.
2. This was not properly revise - There is no concrete theoretical framework regarding the factors? The arguments are too shallow, it lacks substance to support the hypothesis. Regarding hypotheses, there is not enough references to support your hypothesis..
3. Still the reference are too old... the author only add one new reference with a year 2018...
4. The paper added this paragraph in the limitation of the study
"This paper and analyses reveal that mandatory OSH-training of business leaders is positively associated with compliance with legal requirements related to the minimum content of OSH-systems. This means that mandatory OSH-training is important for the establishment of robust occupational safety and health management systems.”
But I believe authors should explain and answer in concise manner the questions they posted in their research in the discussion and add more Implication to academic knowledge and practical implications.
Author Response
Response: Thank you for your new review. Please find reponses below.
Thank you so much for revising your paper but I believe, you still need to revise move to enhance the quality of your paper.
- For example, the introduction part, this whole paragraph was copy paste from previous paper: "Non-compliant acts, defined as deviations from safe operating procedures, standards or rules [8, p. 72] can take place both at the individual level, in relation to company internal safety procedures, and at the organisational level, in relation to national occupational 36 health and safety regulations. Dahl and Søberg [9] argue that more research is needed on 37 organizational compliance or non-compliance with safety regulations and the measures 38 that can be taken to improve compliance at the organizational level." Try checking in Turnitin before you submit your revision, to double check.
Response: The paragraph is now re-written.
- This was not properly revise - There is no concrete theoretical framework regarding the factors? The arguments are too shallow, it lacks substance to support the hypothesis. Regarding hypotheses, there is not enough references to support your hypothesis..
Response: Thank you again for your comment. We have added new literature to support the hypothesis in paragraph 2, section 2.1. With regard to the factors, it is important to note that the factors applied in the current study (as presented in Table 2), derives from checklists applied by the Norwegian Labour Inspection Authority (NLIA). This means that compliance is understood as fulfilling the requirements controlled by the regulatory authority and not as fulfilling factors applied in a theoretical framework.
- Still the reference are too old... the author only add one new reference with a year 2018...
Response:
Thank you for this comment. Actually, six newer references have already been added based on the first review, numbered 2-7 in the paper. As you can see, all of these are from 2018 and newer:
G.; Kanse, L.; Hodkiewicz, M.; Parkes, K. A new look at compliance with work procedures: An engagement perspective. Safety science 2018, 105, 46-54.
Bye, R.J.; Aalberg, A.L. Why do they violate the procedures?–An exploratory study within the maritime transportation industry. Safety science 2020, 123, 104538.
Ijaola, I.A.; Omolayo, O.H.; Akerele, A.O.; Osas, E.F.; Sonibare, S.A. Perceived Implications of Non-Compliance with Safety Practices in Construction Projects: Construction Professionals’ Awareness Level. International Journal of Real Estate Studies 2021, 15, 16-26.
Ojuola, J.; Mostafa, S.; Mohamed, S. Investigating the role of leadership in safety outcomes within oil and gas organisations. In Safety and Reliability, Taylor & Francis: 2020; Vol. 39, pp. 121-133.
Kanse, L.; Parkes, K.; Hodkiewicz, M.; Hu, X.; Griffin, M. Are you sure you want me to follow this? A study of procedure management, user perceptions and compliance behaviour. Safety science 2018, 101, 19-32.
After the last review we have now included approximately 10 more/new references in addition, hopefully satisfying this comment. Please see reference list.
- The paper added this paragraph in the limitation of the study
"This paper and analyses reveal that mandatory OSH-training of business leaders is positively associated with compliance with legal requirements related to the minimum content of OSH-systems. This means that mandatory OSH-training is important for the establishment of robust occupational safety and health management systems.”
But I believe authors should explain and answer in concise manner the questions they posted in their research in the discussion and add more Implication to academic knowledge and practical implications.
Response: Thank you for addressing these issues. We have not done several more substantial changes in the discussion, adding also this following paragraph. We would also like to emphasize that particularly the second and last paragraph in part 5 address implication when discussing training. Training perspectives are reflecting practical implications.
“The results from study 2 complements the results from study 1 and 2. While studies 1 and 2 demonstrate that managerial OSH-training is associated with increased com-pliance across organisations, study 2 indicates that the compliance level is relatively stable when managers have conducted OSH-training at times 1 and 2. This finding im-plies that repeated managerial OHS-training potentially will have little effect on com-pliance. Also, study 2 indicates that the managerial OSH-training should have different formats over time to benefit the organisations potentially. Other training interventions could for instance, focus more on creating improvement related to safety behaviour and safety culture in organisations [35]. Another issue is the complexity of measures and the length of improvement programmes. Empirical research illustrates the difficulties of improving safety culture quickly and that safety intervention could build on more re-fined theoretical models [36]. These issues should also be considered when developing OSH-measures in organisations. It is possible to focus both on bottom-up and top-down processes when improving OHS [35], and designers of safety programmes should con-sider both approaches. Another issue is not to consider OSH in isolation but to consider human resource perspectives in parallel. Research indicates that multiple general issues are related to OSH and essential to consider, such as remote work during the pandemic [37], work climate [38], psychological needs of workers [39], change processes [40], cognitive functions [41] and meaning of work [42]. These topics are essential in addition to managerial OSH-training.”
We hope you are satisfied with the improvements. Thank you for taking the time to comment and improve the paper. We wish you all the best.